# Features of Congenital Arthrogryposis Due to Abnormalities in Collagen Homeostasis, a Scoping Review

**DOI:** 10.3390/ijms241713545

**Published:** 2023-08-31

**Authors:** Sarah MacKenzie Picker, George Parker, Paul Gissen

**Affiliations:** 1Northumbria Healthcare Trust, North Shields NE29 8NH, UK; sarah.picker98@gmail.com; 2Newcastle University Medical School, Newcastle NE2 4HH, UK; georgeparker6199@gmail.com; 3National Institute for Health Research Great Ormond Street Hospital Biomedical Research Centre, University College London, London WC1N 1EH, UK; 4Genetics and Genomic Medicine Department, Great Ormond Street Institute of Child Health, University College London, London WC1N 1EH, UK

**Keywords:** congenital arthrogryposis, collagen, contractures, phenotype, genotype

## Abstract

Congenital arthrogryposis (CA) refers to the presence of multiple contractures at birth. It is a feature of several inherited syndromes, notable amongst them are disorders of collagen formation. This review aims to characterize disorders that directly or indirectly impact collagen structure and function leading to CA in search for common phenotypic or pathophysiological features, possible genotype–phenotype correlation, and potential novel treatment approaches based on a better understanding of the underlying pathomechanism. Nine genes, corresponding to five clinical phenotypes, were identified after a literature search. The most notable trend was the extreme phenotype variability. Clinical features across all syndromes ranged from subtle with minimal congenital contractures, to severe with multiple congenital contractures and extra-articular features including skin, respiratory, or other manifestations. Five of the identified genes were involved in the function of the Lysyl Hydroxylase 2 or 3 enzymes, which enable the hydroxylation and/or glycosylation of lysyl residues to allow the formation of the collagen superstructure. Whilst current treatment approaches are post-natal surgical correction, there are also potential in-utero therapies being developed. Cyclosporin A showed promise in treating collagen VI disorders although there is an associated risk of immunosuppression. The treatments that could be in the clinical trials soon are the splice correction therapies in collagen VI-related disorders.

## 1. Introduction

A clinical finding of congenital arthrogryposis (CA) refers to the presence of two or more joint contractures present at birth that typically affect more than one limb [1,2]. CA is a heterogenous group of over 400 disorders that includes isolated or syndromic forms which, in addition to arthrogryposis, may involve various other organ systems such as kidneys, liver, skin blood cells and others [2]). The common pathway for the pathogenesis of CA involves the failure of normal foetal movement, also referred to as hypokinesia/akinesia [3]. Different aetiologies of the reduction in foetal movements leading to CA can be subdivided into five major categories: neuropathic, myopathic, neuromuscular junction dysfunction, connective tissue abnormalities, and external and environmental factors such as foetal crowding and liquor volume affecting intrauterine position, teratogen exposure, or maternal infection [3]. Whilst it is known that decreased foetal movements lead to the deposition of excess connective tissue around the joints, the underlying pathophysiology of this process is poorly understood. Furthermore, in many of the causative syndromes, knowledge of the precise mechanism by which they result in foetal akinesia is also limited. Of particular importance in this regard are disorders that cause CA through disruption of the normal formation of collagen, of which there are several. 

Connective tissue is embryologically derived from the mesenchyme and largely consists of the following components: cells, elastin fibres, extracellular matrix and collagen family proteins. Structurally, collagens are comprised of characteristic triple-helical units formed of three homotrimeric or heterotrimeric polypeptide chains (Figure 1) [4,5]. Collagens are the most abundant proteins in the human body, accounting for 30% of its dry weight and contributing to the mechanical properties, organisation, structure, and shape of tissues [6]. There are twenty-eight distinct protein types in the collagen family, some of which have restricted tissue distribution and, therefore, a more specific function. This family can be further split into classes; classic fibrillar and network-forming collagens, FACITs (fibril-associated collagens with interrupted triple helices), MACITs (membrane-associated collagens with interrupted triple helices), and MULTIPLEXINs (multiple triple-helix domains and interruptions) [7].

Collagen types I–V are the most common of the twenty-eight types. Type I, II, III and V are classic fibrillar collagens, whereas type IV is a network-forming collagen. Types I and V contribute to the structural formation of bone, tendon, and ligament whereas, type II principally makes up the fibrillar matrix of articular cartilage. Type IV collagen has a crucial role in the structure of basement membranes due to its fibres linking head-to-head rather than in parallel. In addition, the type IV supramolecule lacks the regular glycine in every third residue, which allows it to form a flat sheet [4,8].

Much can be learnt about the common mechanisms of disease by studying phenotypes caused by rare genetic mutations. Thus, in order to gain further insight into possible molecular links between connective tissue defects, foetal akinesia, and contracture formation, we reviewed clinical and molecular features of a CA sub-group caused by the mutations in genes encoding collagen family proteins and collagen-related genes. The connective tissue reorganisation is the likely common pathway for the development of secondary arthrogryposis (of any aetiology). Hence our hypothesis was that by defining the primary defects in genes crucial for connective tissue formation and maintenance that lead to arthrogryposis, we may be able to highlight the potential novel treatment avenues. Several previous studies have investigated molecular mechanisms underlying the cell and tissue abnormalities caused by defects in individual collagens or collagen-related genes. However, there is little understanding as to why some of the collagen defects are associated with CA and others do not. There has also been limited examination of these clinical syndromes as a cohort which may reveal commonalities in their pathogenesis. Therefore, in this review, we intend to examine the following questions:Are there common phenotypic features of the CA caused by collagen defects?Are there mechanistic similarities in collagen formation or modification defects that lead to CA?Could collagen or collagen-related genes be potential targets for CA therapy development?

Hall and Kiefer provided an extensive review of different types of CA [9]. From this we identified the eight initial genes which are described in Table 1. To complement the information provided in the ontology, we performed a literature search in order to produce this scoping review. The full criteria of this search are described in the Appendix A. In brief, the keywords used in PubMed were; “(FKBP65 OR PLOD2 OR PLOD3 OR COLL11A2 OR COL6A1 OR COL6A2 OR COL6A3 OR VPS33B OR VIPAS39) AND (arthrogryposis OR amyloplasia OR “congenital contractures”). A total of 60 articles underwent a full-text review. To be included, the papers were required to be in English, the contractures mentioned had to be congenital, and the paper had to have collagen in the context of arthrogryposis (see Appendix A). Of these 60 articles, 26 met our inclusion criteria—34 articles being excluded. Articles included in this study were published between 1998 and 2022. Several additional papers were identified through the references of the initial manuscripts. From these additional papers, the *COL25A1* gene was also identified as relevant, bringing the total to nine genes (Table 1). Herein, we describe the molecular, cellular, and biochemical features of these genes and encoded proteins as well as the clinical features of the CA resulting from germline mutations.

## 2. Inherited Defects in Collagen and Collagen-Related Genes 

### 2.1. Disorders Related to Collagen Defects

#### Collagen VI Related Diseases

Collagen VI-related myopathies are a group of connective tissue disorders with highly variable phenotypic characteristics that commonly features CA. They include Bethlem myopathy, Ulrich congenital muscular dystrophy (UCMD) and intermediate collagen VI-related myopathies [10]. Bethlem myopathy tends to be milder and is characterized by proximal muscle weakness and distal contractures [11]. These usually develop within the first or second decades of life and progress slowly [11]. Less commonly, contractures may be present congenitally [11]. UCMD is more severe and usually involves early or congenital proximal contractures and striking distal hyperlaxity of joints [11,12]. Independent ambulation is unlikely beyond the first decade and diaphragmatic weakness often results in respiratory failure [11]. The distinction between the two syndromes is often vague and difficult both genetically and phenotypically; hence, the proposed third category is termed “intermediate collagen VI-related myopathies” [10]. All are caused by mutations in the genes *COL6A1, COL6A2,* or *COL6A3*, which encode the three α alpha chains (α1(VI), α2(VI), and α3(VI), respectively) that make up the collagen VI heterotrimer [13]. A further three novel chains, α4(VI), α5(VI), and α6(VI) have recently been identified and these are encoded by the genes *COL6A4, COL6A5,* and *COL6A6*, respectively, but no pathogenic variants associated with these genes have yet been identified [13]. 

The examination of genotype-phenotype correlation related to mutations in genes encoding α(VI) chains, shows great variability in the phenotypes, mutations and mode of inheritance. Although UCMD was initially thought to be caused by recessive mutations, it is now evident that de novo autosomal dominant mutations are its most common cause [14,15,16]. Often, mutations that lead to in-frame skipping of exons in the N-terminal domain of one of the three main α(VI) chains are seen [14,15,17]. These lead to incorporation of malformed α-chain into the triple helix monomer which can then be incorporated into the collagen VI tetramer [17]. This results in a dominant-negative effect as it is highly likely that most collagen VI tetramers will have at least one monomer containing the mutated α(VI) chain [14]. Omission of exon 16 in the α3(VI) chain is the most common mutation of this type and is usually associated with severe UCMD [14,15,18]. Indeed, heterozygous deletion of exon 16 in mice results in a severe UCMD-like clinical phenotype [19] Homozygous missense mutations can also produce UCMD phenotype, but this is much less common and the effect on phenotype of a given missense mutation is hard to predict [15]. Similarly, BM can be inherited in both a dominant and recessive pattern, though the majority are autosomal dominant [20,21,22,23,24,25]. The most common mutation type is skipping of exon 14 in the α1(VI) chain and a BM phenotype can be induced in zebrafish in which skipping of exon 14 is provoked [15,26,27,28,29]. 

Since congenital contractures are much more common in UCMD, the known mutations that produce this more severe phenotype, with either non-functional collagen VI in de novo autosomal dominant mutations or absent collagen VI in homozygous recessive mutations, this allows some genotype-phenotype correlation. Establishing a clearer picture of genotype-phenotype correlation is important when considering future treatment approaches. Bolduc et al. demonstrated that small interfering RNA that specifically targeted common mutations that resulted in skipping exon 16 in *COL6A3* could improve the quantity and quality of the collagen VI matrix produced by UCMD fibroblasts [30]. 

Formation of collagen VI occurs intracellularly from a 1:1:1 ratio of alpha chains 1–3 [31]. These are first assembled as antiparallel dimers and then subsequently tetramers stabilized by disulfide bonds [13]. In the extracellular space, the tetramers form non-covalent bonds to produce the final microfilament structure found in extracellular matrices [11]. Collagen VI has many characterized functions including structural and mechanical roles as are typical of collagens, but also inhibition of apoptosis, regulation of the autophagic process, cellular adhesion, and cellular proliferation [11]. Key roles have been suggested for it in several tissues, including skeletal muscle, tendon, skin, bone, and cartilage. The influence of defective collagen VI function on the development of congenital contractures appears most related to its effects within skeletal muscles and tendons. 

Collagen VI is diffusely expressed in skeletal muscle where it forms a key component of the extracellular matrix across all muscle layers [32]. Its deposition is largely controlled by interstitial fibroblasts though quiescent satellite cells also express it prior to activation [32,33]. 

Mice null in collagen VI through homozygous knockout of the *COL6A1* gene display histological features of myopathy including muscle fibre necrosis and phagocytosis [34]. The absence of collagen VI results in failure of repair processes following injury, defective autophagy, mitochondrial dysfunction, and cell death [11,33,35,36]. Mitochondrial dysfunction appears to be the key to pathogenesis; murine studies demonstrated defective mitochondrial permeability transition pore (PTP) function [36]. Mitochondrial PTP is cyclosporin A (CsA) sensitive and administration of CsA in *COL6A1^−/−^* mice treated the myopathy in this mouse model [36]. CsA also proved promising in a pilot trial of five patients, though its strong immunosuppressive effects are concerning [37]. It is also worth noting that this treatment was given to patients with a minimum age of 6 years so could not prevent congenital contracture formation [37]. Testing of further potential treatments on murine models is limited by the mild BM-like phenotype mice display in the absence of collagen VI compared to humans with UCMD. As a consequence, alternative animal models have been sought [36]. Zebrafish, in particular, have been investigated. The induction of mutations in exon 9 of *COL6A1* in Zebrafish results in a more severe and early onset myopathy than that observed in mice, as well as mitochondrial dysfunction and cell death [38]. This phenotype is more similar to that of UCMD observed in humans. The induction of a mutation in exon 13 of *COL6A1* can also be used to produce a milder phenotype similar to BM [38]. Both CsA and a novel cyclophilin inhibitor, termed NIM-811 have proven successful in improving the induced UCMD-like myopathy in Zebra fish [38,39]. NIM-811 results from a slight structural modification of CsA that exhibits a highly significant decrease in immunosuppressive activity whilst retaining its ability to inhibit the opening of mitochondrial PTPs [39,40]. NIM811 was shown to be significantly more effective than CsA, being able to greatly reduce structural and functional abnormalities if administered relatively rapidly following fertilization [39]. 

In concordance with mouse and zebrafish models, phenotypic analysis has demonstrated defective function of human mitochondrial PTPs [41]. This is evidenced by PTP-dependent depolarisation of mitochondria on the addition of oligomycin or rotenone to cultures of myoblasts or fibroblasts from muscle biopsies of patients with Collagen VI myopathies [42,43]. As with animal models, this effect is prevented with the addition of CsA or NIM-811 [37,39,43]. 

The alterations in tendon structure and function that result from collagen VI deficiency appear to be a key event in contracture formation. In tendons, collagen IV forms a network of beaded filaments in the ECM where, it interacts with a large number of ECM proteins including fibronectin and has a strong interaction with type IV collagen [44,45]. 

The biomechanical properties of tendons of mice unable to produce collagen VI are also significantly altered. They display a significant decrease in cross-sectional area, maximum load, and maximum stiffness [46]. The microdomains and fibrillogenesis are also abnormal.

In vitro studies further demonstrated alteration in organisation of ECM components and increased expression of metalloproteinase 2 (MMP2), a protein involved in the turnover of tendon matrix and fibril growth [47,48]. This likely represents an attempt to augment the defects resulting from collagen VI deficiency, although the mechanism behind MMP2 expression is complex and remains largely unclear [47]. 

The strong evidence of alterations to the tendon ECM due to collagen VI-related disorders gives weight to the proposed contribution of abnormal tendon structure and function to the congenital contractures often seen in these disorders although the precise mechanism through which this causes or contributes to contracture development is uncertain [48]. 

### 2.2. Collagen XXV Related Diseases

Natera-de Benito et al., in 2022 reported five patients from three families with homozygous or compound heterozygous variations in *COL25A1* that had congenital contractures [3]. Phenotype was highly variable, even intrafamilially and the contractures ranged from being mild with only distal upper limb involvement to severe with multiple contractures in one patient [3]. The most common features were finger and knee contractures, as well as disorders in ocular motility and respiratory muscle involvement [3]. The seemingly limited role of collagen XXV in connective tissue development and its key role in skeletal muscle innervation would suggest that improper innervation of skeletal muscle resulted in subsequent fibrosis and foetal akinesia. This is supported by the myopathic features seen on muscle biopsy of these individuals, though no abnormalities were seen on EMG data [3]. The authors speculate that this was due to the high variability in the innervation defect. Of further note is that previous case reports have presented patients with pathogenic variations in the *COL25A1* gene, however, no individual in these studies had joint contractures [49,50]. Natera-de Benito et al. noted that their observed patients had no further variants in potentially pathogenic genes other than those in *COL25A1*, nor did their patients variations appear likely to have a greater effect on collagen XXV function or expression level [3]. The variability between the patients suggests that non-genetic factors may be responsible for the phenotype diversity. 

Collagen XXV is a membrane-bound MACIT collagen, consisting of three collagenous triple-helical domains and four short non-collagenous domains. It is involved in early myogenesis during the formation of primary myofibers [51]. Developing limb skeletal muscles show high levels of collagen XXV expression that regulates the complicated process underlying intramuscular motor innervation through its interaction with the receptor protein tyrosine phosphatases (PTPs) σ and δ [52,53]. PTPσ/δ mediates the interaction of axon-to-muscle cell contact through the binding of the collagen XXV substrate in the early stages of myogenesis [3]. Tanaka et al. demonstrated that *COL25A1* knockout mice die immediately after birth of respiratory failure [54]. Examination of these mice subsequently revealed greatly decreased motoneurons in the spinal cord and abnormal morphology of those that were present [54]. Motor axon bundles were demonstrated to successfully reach the target muscles but then failed to form intramuscular branches and subsequent axonal degeneration was observed [54]. This resulted in motoneuron apoptosis [54]. In adult mammals, collagen XXV mRNA expression is entirely limited to neurons [54]. This evidence indicates the role of collagen XXV in early skeletal myogenesis and suggests a pathogenic mechanism for the neuromuscular disorders observed in humans with homozygous or compound heterozygous variations in *COL25A1* [49,50]. 

### 2.3. Disorders Related to Collagen Modifying Defects

#### LH2 and FKBP65 Related Diseases

Bruck syndrome 2 (BS2) refers to a connective tissue disorder that results from mutations in *PLOD2* that interfere with the ability of LH2 to hydroxylate lysyl residues on the telopeptide. LH2 is encoded by the *PLOD2* gene which encodes two different splice variants: LH2a and LH2b. They differ by 21 amino acid residues encoded on exon 13A [55]. LH2b is more widely expressed, particularly in fibrillar collagen-rich tissues and is the variant required for telopeptide lysyl residue hydroxylation. BS2 is characterized clinically by recurrent fractures, congenital contractures, skeletal deformities, blue sclera, and hearing loss [56]. An identical clinical phenotype called Bruck syndrome 1 (BS1) is also observed in individuals with functional LH2 but mutations in the *FKBP10* gene [57]. *FKBP10* encodes the collagen-specific chaperone protein FKBP65 [58]. It was recently demonstrated to form a complex with LH2 in the ER that is required for LH2 dimerization and telopeptide hydroxylase activity and is therefore also essential to the formation of pyridinolines [59]. 

The most well-characterised feature of BS1 and BS2 is the severe decrease in hydroxylation of lysyl residues in bone type I collagens and a corresponding decrease in pyridinolines cross-links [58]. This finding has been repeatedly observed in individuals with mutations in either of *FKBP10* and *PLOD2*, and electrophoresis of bone type I collagen extracted from affected individuals shows a cross-link pattern that resembles skin [58,60]. No change in hydroxylation of lysyl residues at helical collagen domain sites is observed. Notably, the proportion of hydroxylated lysyl residues is normal in cartilage type II collagen [60]. A recently developed zebrafish model of Bruck syndrome also showed near absent lysyl hydroxylation in bone type I collagen and a disturbance in the bone collagen fibrillar architecture of these fish was also seen [61]. Given the high proportion of hydroxylated lysyl residues usually found in tendon and the propensity of tendon abnormalities to cause congenital contractures, it seems reasonable to suggest that altered cross-linking chemistry in tendon could be a significant factor in contracture development [62]. Indeed, the zebrafish model of Bruck syndrome did report that the fish have a reduced diameter of muscle fibers at the horizontal myoseptum, a structure that is functionally a tendon in zebrafish [61]. They also reported enlargement of a connective tissue layer ensheating muscle fibers known as the endomysium [61]. 

The clinical features appear to be identical for both BS1 and BS2 thus emphasizing the synergistic function of LH2 and FKBP65 [57]. In a case series by Otaify et al. four patients were identified with the c.807_828dup (p.P277AfsX103) in the *FKBP10* gene in two different families [57]. There was significant phenotype variability, with one patient having no contractures at all and another having the very rarely occurring microcephaly [57]. Leal et al. also report a case of two siblings with identical compound heterozygous mutations in *PLOD2* where one sibling had normally formed joints and the other had multiple severe contractures [62]. It is interesting to note that over half of mutations identified occur within the DNA residues of *PLOD2* encoding amino acids 610 to 629 of LH2, suggesting that this domain that forms a flexible loop has a functional importance [57,63].

### 2.4. VPS33B, VIPAR and LHS Related Diseases

Arthrogryposis, renal dysfunction and cholestasis (ARC) is a multisystem disorder with a spectrum of phenotypes, most severe leading to death from overwhelming infection or severe bleeding before the age of two years [64]. 75% of ARC cases are caused by germline mutations in *VPS33B* and 25% of cases are due to germline mutations in *VIPAS*39 encoding VPS33B and VIPAR respectively [64]. *VPS33B* is a gene located at C15q26.1 locus [65]. A wide variety of homozygous or compound heterozygous mutations having been reported. It is interesting to note the genotype-phenotype correlation in *VPS33B* related disease [66,67,68,69]. Smith et al. reported two patients with a mild form of ARC who did have CA. Both displayed a novel *VPS33B* donor splice-site mutation c.1255+5G>C which resulted in partial preservation of protein function in the CHEVI complex. These patients are still alive aged >15, display progressive liver disease and variable degree of arthrogryposis (66 and Gissen personal communication). Genotype-phenotype correlation is also identifiable in patients homozygous or compound heterozygous for the missense mutation Gly131Glu in *VPS33B* [70,71]. This genotype results in a milder phenotype reported as autosomal recessive keratoderma-ichthyosis-deafness (ARKID) syndrome without contractures. 

VPS33B and VIPAR form a stable protein complex termed the class C Homologues in Endosome–Vesicle Interaction (CHEVI) that functions as a discrete endosomal tethering complex in multiple systems. 

The importance of the CHEVI complex in intracellular transport has been demonstrated in a range of tissues such as skin keratinocytes and blood megakaryocytes. 

CHEVI is also essential for the trafficking of lysyl hydroxylase isoenzyme 3 (LH3) from the Golgi-transfer network to the compartments named Collagen IV Carriers, enabling the post-translational modification of collagen IV in the inner medullary collecting duct cells of mice [72]. This trafficking occurs through a Rab10/Rab25-dependent pathway (Figure 2) [72]. The importance of this intracellular trafficking has been demonstrated through the abnormal post-translational modification of collagen IV by LH3 in a murine kidney cell line when deficient in VIPAR or VPS33B [72]. LH3 is essential for collagen IV biosynthesis and basement membrane stability, its absence is fatal in mice embryos [73]. Collagen I is also affected by deficiencies in VPS33B and VIPAR; abnormal collagen I structure was demonstrated in the tail tendons of VPS33B and VIPAR knockout mice [72]. The phenotypic effect of VPS33B or VIPAR mutations in patients with ARC has been demonstrated by the accumulation of procollagen I in fibroblasts cultured in vitro from these patients [72]. The LH3-dependent post-translational modification of collagen was also found to be reduced in the urine of ARC patients [72]. The above suggests that it is likely that the modification of multiple collagens including collagen VI could be affected in ARC linking the common features of these disorders.

Although arthrogryposis can result from the incorrect trafficking of LH3 by CHEVI, mutations in *PLOD3* itself, the gene that encodes LH3, can also result in an arthrogryposis phenotype without affecting the CHEVI complex [72,74]. *PLOD3* is found on c7q22.1 and homozygous mutations in it cause a rarely reported connective tissue disease that has some overlap in features with Sticklers syndrome, Ehlers Danlos syndrome and epidermolysis bullosa [74,75,76]. These features include contractures of the fingers predominantly, ocular abnormalities, sensorineural hearing loss, congenital heart defects, fragile skin and facial abnormalities [69,77,78]. Reduced expression of LH3, reduced glycolisation of urinary collagen, and reduced expression of collagen VII have been noted in patients.

LH3 is an enzyme that has three catalytic activities: lysyl hydroxylase (LH), galactosyltransferase (GT), and glucosyltransferase (GGT) [79]. LH enables the hydroxylation of lysyl residues to hydroxylysine and their subsequent glycosylation is catalysed by GT or GGT [80]. This latter function is unique to LH3 alone and as discussed, mutations in LH3 result in widespread connective tissue disorders, demonstrating the importance of LH3 in collagen formation and function. Importantly, blocking only the LH activity of LH3 in mice results in some disruption of basement membrane and collagen fibril organisation but blocking only the GGT activity is lethal embryologically and prevents collagen IV localisation [81]. It has been shown to be crucial for functioning of collagen I, IV, and VI but is involved in the post-translational modification of most collagens [82,83]. 

## 3. Conclusions

In this review, we aimed to identify commonalities between CA syndromes affected by defects in genes involved in collagen homeostasis. We found that each syndrome displayed great variability of presentation in the severity and location of joint contractures. For example, the contractures in ARC syndrome could be limited to a single joint such as talipes equinovarus or extend to severe hip dislocation in combination with multiple joint contractures. Similarly, there was great variability in the non-arthrogryposis manifestations of these conditions, which is likely to stem from the broad range of functions that collagens have, particularly those that are non-structural. This is especially notable in the collagen VI disorders; they have diverse non-structural functions in connective tissues and results in an equally diverse phenotypic spectrum. It is interesting, that most disorders have additional skin manifestations such as dry skin in both ARC syndrome and collagen VI disorders, and fragile skin in patients with *PLOD2, FKBP10* and *PLOD3* mutations. Although collagen XXV disorders appear to share this trait of phenotypic variability, cases so far have not had skin manifestations. 

It is interesting to note that of the nine genes discussed, five of them have an effect on collagen post-translational modification either by directly encoding an LH enzyme or by encoding a protein that is key to the function of one of these enzymes. Both LH2 and LH3 have unique functions, meaning there is no compensation in the event of a loss of function mutation. For LH2, this is the hydroxylation of lysyl residues on telopeptides to form pyridinolines, and in LH3, it is the glycosylation of hydroxylysine residues. To the contrary, there is a redundancy to LH1, as both LH2 and LH3 can perform the same function. Thus, although LH1 dysfunction produces an Ehlers–Danlos syndrome phenotype, similar to that seen in LH2 and LH3 disorders, no cases of congenital contractures have been recorded [84,85]. 

By definition, congenital arthrogryposis is present at birth, and therefore, in order to prevent contractures, intervention would have to be in utero. This may become possible in the future if gene-based therapies such as the proposed CsA for UCMD are made available for foetal administration. Any definitive treatment would also need to be adaptable to multiple congenital contracture syndromes due to the rarity and variability between not only each specific condition but each individual case. Both these criteria were met by Bolduc et al., 2014, in their development of siRNA to correct the collagen VI defects. There are regulatory and financial challenges for developing mutation-specific therapies for very small numbers of patients. Given the extreme genotypic variability of collagen-related disorders that cause arthrogryposis, selecting genetic targets is a great challenge. In the interim, a deeper understanding of why these genotypes produce such variable phenotypes would also allow better pre-partum counselling, particularly with regards to collagen VI abnormalities, which can range from mild to severely life-limiting. In the short term, an improved understanding of the underlying cellular and molecular pathology of arthrogryposis may allow targeting of the connective tissue surrounding the joints to improve mobility and reduce the incidence and severity of contractures.

## Figures and Tables

**Figure 1 ijms-24-13545-f001:**
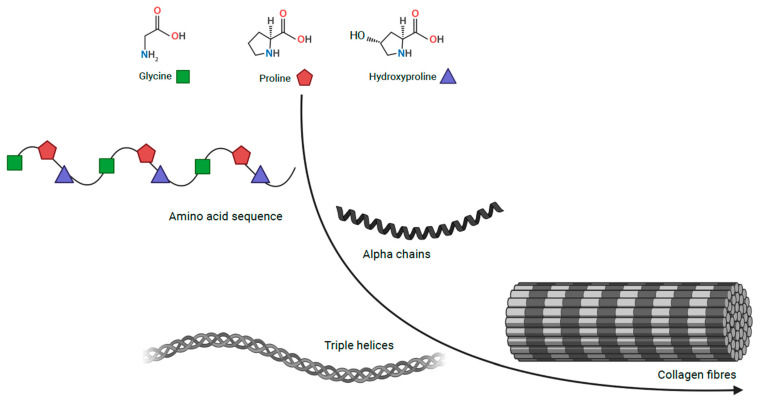
Collagen structure Collagen, comprising of triple helical units formed of three homotrimeric or heterotrimeric polypeptide chains. Coloured shapes are used for glycine in green, proline in red and hydroxyproline in blue.

**Figure 2 ijms-24-13545-f002:**
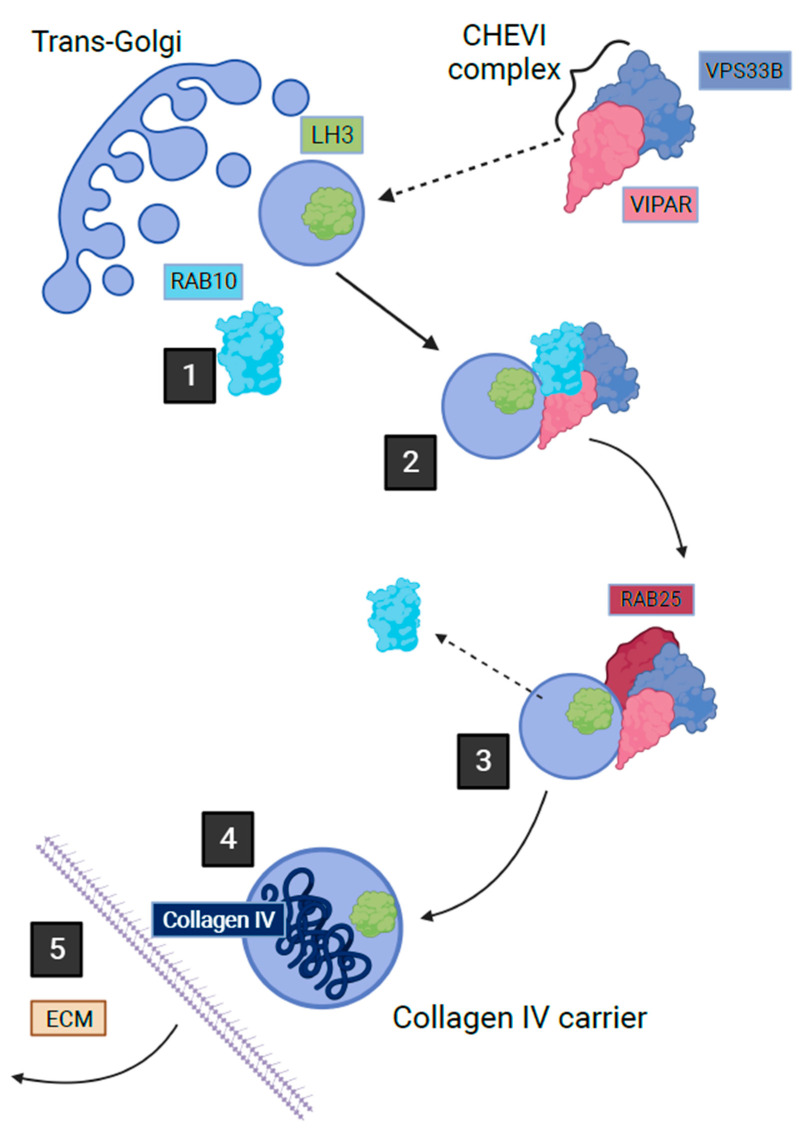
Post-Golgi LH3 trafficking pathway in kidney cells. Step 1 shows how a vesicle containing LH3 buds from Golgi with the help of Rab10 on the cytoplasmic surface. Step 2 shows how the cytoplasmic Chevi complex containing VPS33B and VIPAR proteins assists with the trafficking of the LH3 containing vesicle. Between stages 2 and 3, Rab10 is exchanged with Rab25, which leads LH3 vesicle towards fusion with the Collagen IV-containing organelle where LH3 can proceed with its function in post-translational collagen modification. Stage 4 depicts secretion of Collagen IV carrier outside of the cell (adapted from Banushi et al., 2016 with permission from Paul Gissen).

**Table 1 ijms-24-13545-t001:** Summary of characteristics of genes encoding collagen or collagen-modifying proteins.

Gene	Protein Encoded	Function of Protein	Clinical Phenotype
*PLOD2*	LH2	Post-translational modification of collagen	Congenital contractures, joint stiffness, osteoporosis, recurrent fractures, short stature, pterygia, kyphosis, idiopathic scoliosis, fragile skin
*FKBP10*	FKBP65	Forms a complex with LH2 to allow LH2 dimerization and telopeptide hydroxylase activity	Congenital contractures, joint stiffness, osteoporosis, recurrent fractures, short stature, pterygia, kyphosis, idiopathic scoliosis, aplastic patellae & radius, talipes, reduced tendon reflexes, melanocytic naevus, fragile skin
*COL6A1*	Collagen VI alpha 1 chain	Structural role in basement membrane, inhibition of apoptosis, regulation of cellular adhesion and cellular proliferation	Congenital contractures, muscle weakness, hyperextensibility of joints, kyphosis, spinal rigidity, decreased foetal movements, diaphragmatic weakness, hip dislocation, torticollis, delayed motor skills, keloid scarring, follicular hyperkeratosis
*COL6A2*	Collagen VI alpha 2 chain	Structural role in basement membrane, inhibition of apoptosis, regulation of cellular adhesion and cellular proliferation	Congenital contractures, muscle weakness, hyperextensibility of joints, kyphosis, spinal rigidity, decreased foetal movements, diaphragmatic weakness, hip dislocation, torticollis, delayed motor skills, keloid scarring, follicular hyperkeratosis
*COL6A3*	Collagen VI alpha 3 chain	Structural role in basement membrane, inhibition of apoptosis, regulation of cellular adhesion and cellular proliferation	Congenital contractures, muscle weakness, hyperextensibility of joints, kyphosis, spinal rigidity, decreased foetal movements, diaphragmatic weakness, hip dislocation, torticollis, delayed motor skills, keloid scarring, follicular hyperkeratosis
*COL25A1*	Collagen XXV	Role in myogenesis during the primary formation of myofibrils by enabling motor innervation	Congenital ptosis, Duane syndrome, congenital contractures
*VPS33B*	VPS33B	LH3-mediated post-translational modification of collagen	Congenital contractures, renal tubular dysfunction, cholestasis, ichthyosis
*VIPAS39*	VIPAR	LH3-mediated post-translational modification of collagen	Congenital contractures, renal tubular dysfunction, cholestasis, ichthyosis
*PLOD3*	LH3	Post-translational modification of collagen	Finger contractures, ocular abnormalities, sensorineural hearing loss, congenital heart defects, facial abnormalities, skin blistering

## Data Availability

Data sharing not applicable. No new data were created or analyzed in this study. Data sharing is not applicable to this article.

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
