# Peer review of "Features of Congenital Arthrogryposis Due to Abnormalities in Collagen Homeostasis, a Scoping Review"

_ijms, 2023, doi:10.3390/ijms241713545_

Round 1

Reviewer 1 Report

The provided synthesis is interesting, though it appears more as a dissertation than a review.. To increse the value of the manuscript I would suggest to significantly implement the methodological part, being more specific :

- if it is a scoping review better effort should be done to clarify the link from the studies included into the review and the contents exposed;

- if it is a systematic review the authors should deeply explain the methodological process that was followed, in accordance to the PRISMA checklist.

Furthermore, once defined the type of study I would recommend to declare it into the title.

Author Response

The provided synthesis is interesting, though it appears more as a dissertation than a review.. To increse the value of the manuscript I would suggest to significantly implement the methodological part, being more specific :

- if it is a scoping review better effort should be done to clarify the link from the studies included into the review and the contents exposed;

- if it is a systematic review the authors should deeply explain the methodological process that was followed, in accordance to the PRISMA checklist.

Furthermore, once defined the type of study I would recommend to declare it into the title.

We thank this reviewed for the insightful comments and positive assessment. As suggested by the reviewer we have specified this review being “scoping review” and changed the title accordingly. We have added a supplementary flow chart explaining what methodology was applied.  We also briefly explained it in the text.

Reviewer 2 Report

The review "Features of Congenital Arthrogryposis due to Abnormalities in Collagen Homeostasis" caused me very contradictory feelings. This review summarizes information about the contribution of collagen genes and related genes to the ethiology of arthrogryposis. A lot of material has been collected and carefully presented. However, it is absolutely not clear from the work why this particular group of genes was chosen to study the relationship with congenital atrogripposis. It is known that arthrogryposis can have a neurogenic nature, so it is natural that severe mutations responsible for congenital muscular dystrophy leading to the manifestation of the disease and a decrease in fetal mobility can also lead to contractures.

As far as I understand, the purpose of this work was to designate arthrogryposis as an intrauterine or neonatal marker that allows you to suspect collagenopathy and take a drug that is currently being developed. I believe that the authors should more clearly indicate the purpose of writing the review. Shorten the parts to make it easier to understand. I recommend giving most of the information in tabular form without duplication in the text. The work needs to be compacted and the format of the presentation of the grandiose material is reviewed.

Author Response

The review "Features of Congenital Arthrogryposis due to Abnormalities in Collagen Homeostasis" caused me very contradictory feelings. This review summarizes information about the contribution of collagen genes and related genes to the ethiology of arthrogryposis. A lot of material has been collected and carefully presented.

We thank the reviewer for the positive assessment of our data presentation.

However, it is absolutely not clear from the work why this particular group of genes was chosen to study the relationship with congenital atrogripposis. It is known that arthrogryposis can have a neurogenic nature, so it is natural that severe mutations responsible for congenital muscular dystrophy leading to the manifestation of the disease and a decrease in fetal mobility can also lead to contractures.

We are grateful to this reviewer for noticing that the reason for selecting collagen related genes for this review was not clear enough. This allowed us to highlight that "The connective tissue reorganisation is the likely common pathway for development of secondary arthrogryposis (of any aetiology). Hence our hypothesis was that by defining the primary defects in genes crucial for connective tissue formation and maintenance that lead to arthrogryposis, we may be able to highlight the potential novel treatment avenues".

This has now been emphasised stronger in the text.

As far as I understand, the purpose of this work was to designate arthrogryposis as an intrauterine or neonatal marker that allows you to suspect collagenopathy and take a drug that is currently being developed. I believe that the authors should more clearly indicate the purpose of writing the review. Shorten the parts to make it easier to understand. I recommend giving most of the information in tabular form without duplication in the text. The work needs to be compacted and the format of the presentation of the grandiose material is reviewed.

Thank you for this comment. As stated above our group’s particular interest is in the collagens and collagen related genes defects as key to contracture formation. We feel that the novelty of this review is in combining the molecular, phenotypic and cell biological information which are highlighted as a positive by other reviewers. We have included one additional flow chart but do not feel that another table is needed.

Reviewer 3 Report

In this review, the authors aim to characterize disorders that directly or indirectly impact collagen structure and function leading to Congenital arthrogryposis in search for common phenotypic or pathophysiological features, possible genotype-phenotype correlation, and novel treatment approaches based on better understanding of the underlying pathomechanism. Based on my evaluation, I believe that this work has the potential to make a significant contribution to the field.

To make the manuscript more profound, we suggest adding the following modifications:

1.     For the scoliosis, please clarify whether they are congenital scoliosis or idiopathic scoliosis.

2.     Please discuss in the Introduction the environmental factors that affect congenital arthrogryposis.

3.     Correction of Formatting Errors: the formatting errors in lines 469 and 471. please remove the unnecessary paragraph breaks, ensuring that the manuscript is presented in a clear and coherent manner.

The English language is very well.

Author Response

In this review, the authors aim to characterize disorders that directly or indirectly impact collagen structure and function leading to Congenital arthrogryposis in search for common phenotypic or pathophysiological features, possible genotype-phenotype correlation, and novel treatment approaches based on better understanding of the underlying pathomechanism. Based on my evaluation, I believe that this work has the potential to make a significant contribution to the field.

We thank the reviewer for these positive comments and suggestions.

To make the manuscript more profound, we suggest adding the following modifications:

  1. For the scoliosis, please clarify whether they are congenital scoliosis or idiopathic scoliosis.

Changes made in the table as suggested by the reviewer.

  1. Please discuss in the Introduction the environmental factors that affect congenital arthrogryposis.

These were added to the introduction as suggested by the reviewer.

  1. Correction of Formatting Errors: the formatting errors in lines 469 and 471. please remove the unnecessary paragraph breaks, ensuring that the manuscript is presented in a clear and coherent manner.

We followed the reviewer’s suggestion.